# BARLERIA: AN EFFICIENT TUNING FRAMEWORK FOR REFERRING IMAGE SEGMENTATION

**Yaoming Wang**[1,†]    **Jin Li**[1,†]    **Xiaopeng Zhang**[2✉]    **Bowen Shi**[1]    **Chenglin Li**[1]
**Wenrui Dai**[1✉]    **Hongkai Xiong**[1]    **Qi Tian**[2]
[1]Shanghai Jiao Tong University    [2]Huawei Cloud
{wang_yaoming, deserve_lj, lcl1985, daiwenrui,
xionghongkai}@sjtu.edu.cn; zxphistory@gmail.com,
tian.qi1@huawei.com

## ABSTRACT

Pre-training followed by full fine-tuning has gradually been substituted by Parameter-Efficient Tuning (PET) in the field of computer vision. PET has gained popularity, especially in the context of large-scale models, due to its ability to reduce transfer learning costs and conserve hardware resources. However, existing PET approaches primarily focus on recognition tasks and typically support uni-modal optimization, while neglecting dense prediction tasks and vision language interactions. To address this limitation, we propose a novel PET framework called **B**i-direction**a**l Inte**r**twined Vision **L**anguage Effici**e**nt Tuning for **R**eferring **I**mage Segment**a**tion (**BarLeRIa**), which leverages bi-directional intertwined vision language adapters to fully exploit the frozen pre-trained models' potential in cross-modal dense prediction tasks. In BarLeRIa, two different tuning modules are employed for efficient attention, one for global, and the other for local, along with an intertwined vision language tuning module for efficient modal fusion. Extensive experiments conducted on RIS benchmarks demonstrate the superiority of BarLeRIa over prior PET methods with a significant margin, *i.e.*, achieving an average improvement of 5.6%. Remarkably, without requiring additional training datasets, BarLeRIa even surpasses SOTA full fine-tuning approaches. The code is available at https://github.com/NastrondAd/BarLeRIa.

## 1    INTRODUCTION

In recent years, large-scale models have made significant contributions to advancements in NLP and CV, However, the cost associated with full fine-tuning of large models has become prohibitively expensive. To address this challenge, Parameter-Efficient Tuning (PET) approaches have emerged as a prevalent paradigm (Houlsby et al., 2019; Jie & Deng, 2022; Jia et al., 2022; Wang et al., 2023). By freezing a majority of the pre-trained model and fine-tuning only a small subset of parameters, PET approaches offer high efficiency while maintaining performance comparable to full fine-tuning, and are increasingly favored for language dialogue (Karimi Mahabadi et al., 2021; Sung et al., 2021) as well as visual recognition tasks (Chen et al., 2022b; Jia et al., 2022). Despite these advancements, limited research has explored the effectiveness of PET pipelines for adapting to dense prediction tasks (Ding et al., 2022; Qian et al., 2023) or facilitating cross-modal fusion.

This paper investigates the generalization ability of Parameter-Efficient Tuning (PET) and examines its affordability for a challenging cross-model dense prediction task Referring Image Segmentation (RIS). RIS is a fundamental segmentation task designed to segment target objects from input images based on given text descriptions (Hu et al., 2016). Different from vanilla segmentation tasks, RIS needs to extract not only spatial and semantic information from images, but also key semantics from textual descriptions, and merge them in order to get the correct segmentation results. Previous studies have approached this task by either concatenating textual embeddings with visual features and incorporating vision-language attention mechanisms to facilitate interactions (Yu

---

Corresponding author: Xiaopeng Zhang; Wenrui Dai. [†]: Equal contribution. This work was done when Yaoming Wang interned at Huawei Inc.

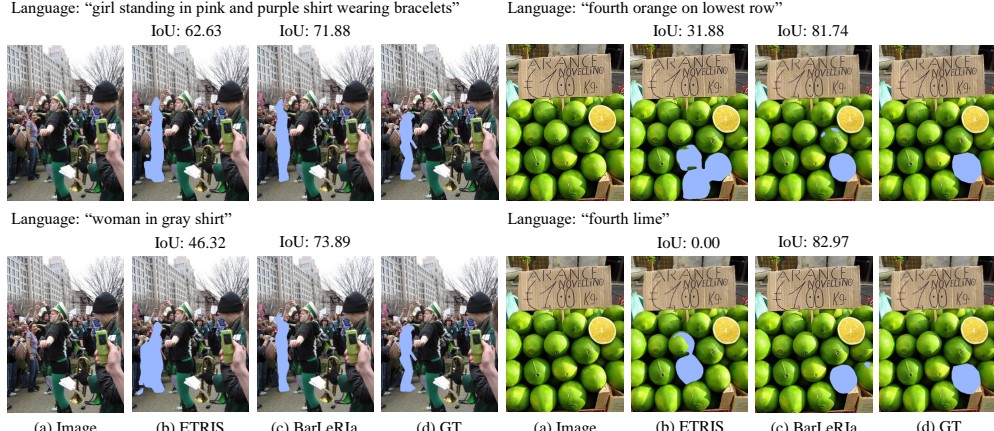

Figure 1: Comparison between BarLeRIa and state-of-the-art PET RIS method ETRIS. We perform experiments using two different referring expressions: detailed or abstracted. In the first row, the expression is detailed and two methods can locate the object given sufficient knowledge, though BarLeRIa outperforms ETRIS a lot. In the second row, only brief expressions are provided, ETRIS locates wrong contours while BarLeRIa still segments the target objects well. Best viewed in color.

et al., 2018; Li et al., 2018; Chen et al., 2019), or by pursuing vision-language alignment using uni-modal pre-trained models supplemented with additional training (Liu et al., 2023; Yan et al., 2023). More recently, leveraging the advancements in vision-language pre-training (Radford et al., 2021), Wang et al. (2022) propose to transfers multi-modal knowledge from CLIP through text-to-pixel contrastive learning, leading to remarkable performance gains. However, these approaches rely on computationally extensive full fine-tuning, which raises concerns about scalability and affordability.

Few works explore integrating PET into RIS task, a pioneering work Xu et al. (2023), introduces a vision-language bridge that combines vision inductive biases and language information, and achieves comparable performance to full fine-tuning. However, this approach primarily focuses on alignment between the vision and language modalities, while overlooking the core aspect of PET, namely, adapting the biased feature from pre-trained models (Jia et al., 2022; Wang et al., 2023). Besides, local modal fusion is adopted in the proposed bridge network as well as the pre-trained vision language models and the segmentation head (Wang et al., 2022). Consequently, all components of the model repetitively fuse local visual features with textual embeddings without incorporating a global prior from the text input to regularize the visual features, which leading to off-target visual information interference and sub-optimal performance.

Considering the above two issues when incorporating PET into RIS task, Firstly, we propose a novel technique to address the feature adaption problem. The highlight is an intertwined vision language efficient tuning framework for better modal fusion along with feature adaption as a basic design. For both visual and textual branches, we fuse the visual and textual input in front of each frozen layer and adapt each layer's shortcut feature distribution via normalizing flow (Wang et al., 2023). In this way, we keep the backbone frozen, employ modal fusion via the original self-attention mechanism, and are able to adapt the biased features for segmentation tasks.

Second, in order to address the global regularization issue, we extract a global prior from the text input to regulate the vision features. This regularization is achieved with a limited number of parameters in an end-to-end manner. Our proposed method consists of a bi-directional efficient tuning framework, which comprises a global prior module and a global tuning network. The global prior module leverages the cosine similarity between visual features and textual embeddings to enforce regularization. Moreover, to ensure that the global prior regularization does not conflict with the local intertwined vision language tuning, we introduce global shortcut tuning modules that are detached from the pre-trained backbone. By doing so, we establish a parallel shortcut tuning network alongside the backbone. Similarly, we extend the intertwined vision language tuning to the shortcut tuning network to facilitate better fusion of the models.

Incorporating the proposed intertwined vision language efficient tuning and the bi-directional efficient tuning modules, we produce a novel PET framework, namely **Bi**-direction**a**l Inte**r**twined Vi-

sion **L**anguage Effici**e**nt Tuning for **R**eferring **I**mage Segmenta**t**ion (**BarLeRIa**), to fully exploit the potential of frozen pre-trained vision-language models. BarLeRIa exhibits remarkable performance improvement with only 0.4% to 2.5% tuning parameters compared with the backbone when utilizing CLIP ViT-B as the pre-trained model. Compared to the state-of-the-art PET approach ETRIS (Xu et al., 2023), BarLeRIa shows a significant improvement, *e.g.*, +2.01 IoU on RefCOCO, +5.19 IoU on RefCOCO+, and +3.74 IoU on G-Ref, respectively. Compared with the fully fine-tuned large visual language model LISA-7B (Lai et al., 2023), BarLeRIa achieves comparable performance with only about 2M learnable backbone parameters and significantly outperforms the untuned 7B model. Besides, BarLeRIa also outperforms full fine-tuning state-of-the-art approaches, *e.g.,* Poly-Former (Liu et al., 2023) and UNINEXT (Yan et al., 2023), which need pre-training on extra region-level datasets. As comparison, without extra pre-training, BarLeRIa achieves a new SOTA performance when adopting EVA-CLIP (Radford et al., 2021) as the pre-trained vision language model.

In a nutshell, our contributions can be summarized as follows:

- We find that previous PET methods for RIS task focus on modal fusion and ignore feature distribution adaptation and propose a novel intertwined vision language efficient tuning algorithm for both feature adaptation and modal fusion with only 0.4M (ViT-B) learnable parameters.
- We reveal that repeating fusing the local visual features with the textual embeddings is another problem for previous approaches and propose a bi-directional efficient tuning framework that enables both local feature fusion and global prior regularization.
- We design a novel global shortcut tuning module that tunes only 1.8M (ViT-B) parameters and learns the global prior regularization in parallel with the backbone to avoid conflicts with our proposed local intertwined vision language efficient tuning.

## 2 METHODOLOGY

### 2.1 PRELIMINARIES

**Adapting Shortcut with Normalizing Flow (SNF)** (Wang et al., 2023) adjusts the shortcut to adapt pre-trained models into downstream tasks. For a given skip connection inside the transformer, it can be depicted as $y = x + f(x)$ where $x$ is the input feature, $f$ is a certain architecture of the transformer and $y$ is the output. During fine-tuning, SNF only operates on the shortcut $x$ while keeping other parts frozen, *i.e.*, $y = s(x) + f(x)$. For a given feature $x \in \mathrm{R}^{N \times d}$, the transformation imposed by SNF is given by:

$$s(x) = x + \lambda \cdot h(\gamma^T \cdot x + \beta) \tag{1}$$

where $\lambda, \gamma, \beta \in \mathcal{R}^d$, $\cdot$ is the Hadamard product and $h(\cdot)$ is a smooth non-parameteric non-linearity. Note that SNF allows for multiple concatenated transformations, i.e., $y = s(s(\cdots s(x))) + f(x)$. The number of transformations is denoted as the depth of SNF.

### 2.2 FRAMEWORK OVERVIEW

The framework of BarLeRIa is depicted in Fig. 2. The fundamental design of BarLeRIa is the proposed intertwined vision language efficient tuning algorithm, which is used to enhance modal fusion. Along with it, we employ a bi-directional efficient tuning framework that simultaneously adjusts local features and extracts global priors from the text input, thereby regularizing the visual features. This framework consists of two distinct efficient tuning modules. The first module, known as the local intertwined module, utilizes the intertwined vision language efficient tuning approach to enable efficient modal fusion and multi-modal feature adaptation. The second module, referred to as the global shortcut tuning module, incorporates a parallel shortcut module and leverages the global prior generated from the global prior module to complement the local vision features. Finally, the complete vision features, alongside the textual embeddings, are inputted into the learnable referring image segmentation head and generate the corresponding segmentation masks.

### 2.3 INTERTWINED VISION LANGUAGE EFFICIENT TUNING

For an input tokenized referring expression $T \in \mathbb{R}^{L \times D}$ and an input tokenized image $I \in \mathbb{R}^{H \times W \times C}$, along with a visual encoder $\phi : \{\phi_1, \cdots, \phi_N\}$ composed of $N$ transformer blocks, we

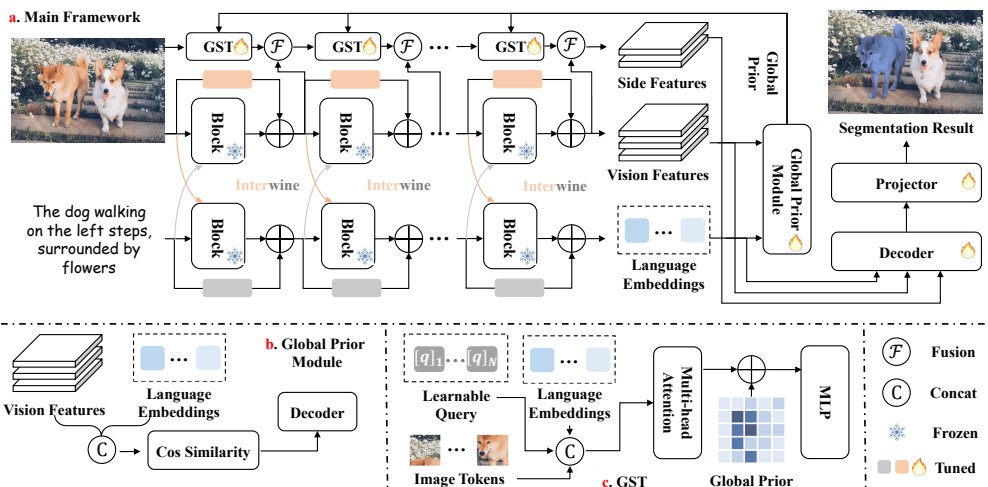

Figure 2: The framework of BarLeRIa. GST is the abbreviation of global shortcut tuning. For the visual branch, we fuse the textual embeddings with the visual input in the frozen visual block and further adapt the feature distribution with the normalizing flow. For the language branch, we concatenate the visual class token to the textual input and achieve modal fusion and feature adaption similarly. Besides, a global shortcut tuning module along with a global prior module is proposed in parallel with the backbone for global visual regularization.

begin by projecting the tokenized expression $T$ as: $T' \leftarrow TW_{proj}$, where $W_{proj} \in \mathbb{R}^{D \times C}$. Next, we concatenate the projected expression $T'$, the tokenized image $I$ (we reshape it into $\mathbb{R}^{(H \cdot W) \times C}$ before), and the class token $cls$ as $[cls, T', I]$. These intertwined embeddings are then fed into the frozen visual encoder $\phi$, and the output is given as:

$$[cls, embed] = \mathcal{F}([cls, T'_{i-1}, embed]) + \phi_i([cls, T'_{i-1}, embed]) \tag{2}$$

where $\mathcal{F} = f_J \circ \cdots \circ f_1$ represents a chain of $J$ invertible feature mappings: $f_i(z) = z + \lambda_i \cdot h(\gamma_i^T \cdot z + \beta_i)$, and $T'_{i-1}$ represents the output from the $(i-1)$-th textual block. Note that we take the projected expression $T'$ as input for the frist layer. With the Multi-Head Self-Attention (MHSA) module employed in each visual block for information interaction between tokens, we successfully achieve modal fusion through these frozen visual blocks. Additionally, the shortcut normalizing flow $\mathcal{F}$ is applied in each visual block for feature adaptation. As for the textual block $\psi_i$, we project the visual class token $cls$ into $cls'$ to align its feature dimension with the textual embeddings $T$. Then, we concatenate the textual embedding $T$ with the projected class token $cls'_{i-1}$ from the previous visual block to form the input. Furthermore, we leverage the shortcut normalizing flow to adapt the shortcut textual embeddings and employ the frozen transformer block to fuse the textual and visual features. Consequently, we obtain the output of the textual block $\psi_i$ as follows:

$$[T] = \mathcal{F}([cls'_{i-1}, T]) + \psi_i([cls'_{i-1}, T]) \tag{3}$$

## 2.4 Bi-directional Efficient Tuning

As discussed in Sec. 1, bi-direction refers to the combination of local efficient tuning and global prior regularization, and the bi-directional efficient tuning framework consists of two modules: the Global Prior Module and the Global Shortcut Tuning Network.

**Global Prior Module.** As language is more semantic-rich, the produced language embeddings tend to be more robust compared to the visual ones. Therefore, we propose regularizing visual features through the global prior generated by the language embeddings. Specifically, given the visual encoder $\phi$ and its output vision feature $F_v$, as well as the language encoder $\psi$ and its output language embeddings $F_l$, we concatenate the vision features with the language embeddings to obtain intertwined features: $[F_l, F_v]$. Next, we calculate the cosine similarity between the intertwined feature $[F_l, F_v]$ and the language embeddings $F_l$. This cosine similarity serves as an attention mask, which is then multiplied with the intertwined feature to produce the global prior $p$:

$$p = \cos([F_l, F_v], F_l) \cdot [F_l, F_v] \tag{4}$$

**Global Shortcut Tuning Network.** To ensure that the global prior regularization does not conflict with our local intertwined vision language efficient tuning and to achieve an end-to-end PET pipeline, we introduce a global shortcut tuning network $\mathcal{G}$ that operates in parallel with the vision encoder. This network consists of $M$ modules $\{\mathcal{G}_1, \cdots, \mathcal{G}_M\}$, each following the design of the transformer block with MHSA and MLP, but with smaller feature dimensions (default setting: 144). Firstly, we transform the global prior $p$ using $M$ linear transformations to obtain adapted priors $p_1, \cdots, p_M$ for each global shortcut tuning module. Then, given the tokenized input image $I$ and the language embedding $F_l$, we concatenate them with learnable query tokens $q$ as $[q, F_l, I]$, which serves as the input for the global shortcut tuning network. For the first global shortcut tuning module $\mathcal{G}_1$, we feed these concatenated tokens along with the global prior $p_1$ as follows:

$$I_1 = \mathcal{G}_1([q, F_l, I], p_1) \tag{5}$$

where $I_1$ represents the output of the first module, and $\mathcal{G}_1(I, p)$ utilizes the global prior $p$ to regularize the MHSA feed-forward in the global shortcut tuning module. In elaboration, we project the input $I$ into query, key, and value tokens using projection matrices $W_q$, $W_k$, and $W_v$, respectively. Additionally, we project the global prior $p$ into complementary value tokens using learnable projection matrices $W_p$. The two sets of value tokens are then added together, resulting in the new value tokens $IW_v + IW_p$. Consequently, we redefine the self-attention mechanism as:

$$\text{MHSA}(I) = \text{Softmax}\left(\frac{IW_q IW_k}{\sqrt{C}}\right)(IW_v + IW_p) \tag{6}$$

Here, $C$ denotes the feature dimension, and for simplicity, we omit the multi-head division. By modifying the value tokens in the MHSA block, we successfully incorporate global prior regularization into the global shortcut tuning module through the introduced global attention. For the remaining $M-1$ modules, we repeat the global prior regularization process as follows:

$$I_i = \mathcal{G}_i(I_{i-1}, p_i), i = 2, \cdots, M \tag{7}$$

The global shortcut tuning network comprises very few parameters and is primarily employed for parameter-efficient tuning. Subsequently, we proceed to fuse the output shortcut features $I_1, \cdots, I_M$ from each global shortcut tuning module with the corresponding vision features $F_v$. For a given $i$-th output shortcut feature $I_i$ and its corresponding vision features $F_v^i$, we first interpolate $I_i$ to match the height and width of the original vision features $F_v^i$. Subsequently, we add these two sets of vision features to obtain the output feature $F_{out}$ as follows:

$$F_{out}^i = \text{Interpolate}(I_i) + F_v^i \tag{8}$$

To prevent conflicts with the local intertwined module, we detach $F_v^i$ during fine-tuning.

## 2.5 FINAL OBJECTIVE

Following Wang et al. (2022) and Xu et al. (2023), we incorporate a learnable referring image segmentation head composed of a cross-modal neck, vision-language decoder, and an up-sample projector to extract the cross-modal intertwined feature $F_{ci}$ and the transformed textual feature $F_t$:

$$F_{ci}, F_t = \text{Head}(F_{out}^M, F_v, F_l) \tag{9}$$

where $F_{out}^M$ represents the output from the last global shortcut tuning module, while $F_l$ and $F_v$ denote the textual embeddings and the vision encoder features adapted by the local intertwined modules. To train our model, we employ a text-to-pixel contrastive loss (Wang et al., 2022) as our training objective, which encourages the alignment of textual embeddings with their corresponding visual pixels, while pushing textual embeddings away from other irrelevant visual pixels. The text-to-pixel contrastive loss is formulated as follows:

$$\mathcal{L}_{\text{ttp}}^i\left(F_{ci}^i, F_t\right) = \begin{cases} -\log\left(\sigma\left(F_{ci}^i \cdot F_t\right)\right), & i \in \mathcal{P} \\ -\log\left(1 - \sigma\left(F_{ci}^i \cdot F_t\right)\right), & i \in \mathcal{N} \end{cases}$$
$$\mathcal{L}_{\text{ttp}}\left(F_{ci}, F_t\right) = \frac{1}{|\mathcal{P} \cup \mathcal{N}|} \sum_{i \in \mathcal{P} \cup \mathcal{N}} \mathcal{L}_{\text{ttp}}^i\left(F_{ci}^i, F_t\right) \tag{10}$$

where $\sigma$ denotes the sigmoid function, $\mathcal{P}$ and $\mathcal{N}$ represent the classes of 1 and 0, respectively.

Table 1: Comparison with SOTA RIS methods and the PET RIS SOTA method without additional datasets evaluated using the IoU metric on RefCOCO-related datasets.

| Method | RefCOCO | | | RefCOCO+ | | | G-Ref | | | Avg |
|---|---|---|---|---|---|---|---|---|---|---|
| | val | testA | testB | val | testA | testB | val(u) | test(u) | val(g) | |
| Traditional Full Fine-tuning | | | | | | | | | | |
| MAttNet (Yu et al., 2018) | 56.5 | 62.4 | 51.7 | 46.7 | 52.4 | 40.1 | 47.6 | 48.6 | - | 50.5 |
| RRN (Li et al., 2018) | 55.3 | 57.3 | 54.0 | 39.8 | 42.2 | 36.1 | - | - | 36.5 | 43.8 |
| CMSA (Ye et al., 2019) | 58.3 | 60.6 | 55.1 | 43.8 | 47.6 | 37.9 | - | - | 40.0 | 47.0 |
| CAC (Chen et al., 2019) | 58.9 | 61.8 | 53.8 | - | - | - | 46.4 | 47.0 | 44.3 | - |
| BRINet (Hu et al., 2020) | 61.4 | 63.4 | 59.6 | 48.6 | 52.9 | 42.1 | - | - | 48.0 | 52.5 |
| CMPC+ (Liu et al., 2021a) | 61.4 | 64.5 | 59.6 | 49.6 | 53.4 | 43.2 | - | - | - | - |
| CGAN (Luo et al., 2020) | 64.9 | 68.0 | 62.1 | 51.0 | 55.5 | 44.1 | 51.0 | 51.7 | - | 55.5 |
| LTS (Jing et al., 2021) | 65.4 | 67.8 | 63.1 | 54.2 | 58.3 | 48.0 | - | - | - | - |
| VLT (Ding et al., 2021) | 65.7 | 68.3 | 62.7 | 55.5 | 59.2 | 49.4 | - | - | 49.8 | 56.7 |
| PCAN (Chen et al., 2022a) | 69.5 | 71.6 | 64.2 | 58.3 | 63.7 | 48.9 | 60.0 | 60.8 | 57.5 | 61.6 |
| ReSTR (Kim et al., 2022) | 67.2 | 69.3 | 64.5 | 55.8 | 60.4 | 48.3 | 54.5 | - | 54.5 | 58.8 |
| CRIS (Wang et al., 2022) | 70.5 | 73.2 | 66.1 | 62.3 | 68.1 | 53.7 | 59.9 | 60.4 | - | 63.8 |
| LAVT (Yang et al., 2021) | 72.7 | 75.8 | **68.8** | 62.1 | 68.4 | 55.1 | - | - | 60.5 | 64.9 |
| WiCo (Cheng et al., 2023) | **73.5** | **76.9** | 68.1 | 63.4 | 69.2 | 55.8 | - | - | 60.2 | 65.3 |
| Parameter Efficient-Tuning | | | | | | | | | | |
| ETRIS (Xu et al., 2023) | 70.5 | 73.5 | 66.6 | 60.1 | 66.9 | 50.2 | 59.8 | 59.9 | 57.9 | 62.8 |
| Ours | 72.4 | 75.9 | 68.3 | **65.0** | **70.8** | **56.9** | **63.4** | **63.8** | **61.6** | **66.5** |

## 3 EXPERIMENTS

### 3.1 EXPERIMENTAL SETUP

**Datasets.** We employ three challenging referring image segmentation benchmarks in our experiments: RefCOCO (Kazemzadeh et al., 2014), RefCOCO+ (Kazemzadeh et al., 2014), and G-Ref (Yu et al., 2016). Please refer to the appendix A.1 for details.

**Implementation Details.** We train the whole network in an end-to-end manner for 50 epochs using the Adam optimizer with a learning rate of 0.0001. A learning rate decay is employed at the 35th epoch with a decay factor of 0.1. We train the model using 2 Tesla V100 GPUs with a batch size of 32. For ViT-L/14, we train the model using 8 Tesla V100 GPUs with a batch size of 64 and an initial learning rate of 0.0002. Following previous works (Ding et al., 2021; Liu et al., 2017; Wang et al., 2022; Xu et al., 2023), we adopt IoU as the metric to evaluate the performance. More details can be referred to in the appendix A.2.

### 3.2 MAIN RESULTS

We conducted a comprehensive comparison between our BarLeRIa and a series of previous RIS approaches. The results, presented in Tab. 1, demonstrate that our approach significantly outperforms state-of-the-art RIS methods on three commonly used datasets, achieving 6 SOTA and 3 sub-sub-SOTA performance across 9 evaluation tasks. In particular, we surpass the performance of WiCo (Cheng et al., 2023), which utilizes an additional ResNet-50 to extract top-down segmentation proposals as a pre-stage. In contrast, our BarLeRIa model achieves superior results using only 2.2M parameters and is trained in an end-to-end manner.

Furthermore, we compare our proposed approach with the state-of-the-art parameter-efficient tuning RIS method, ETRIS (Xu et al., 2023). To ensure a fair comparison, we employ the same CLIP pre-trained vision language model of ViT-B/16 as used in ETRIS and freeze the visual and textual encoders. The tuning backbone parameters in BarLeRIa are only 2.2M, which is comparable to ETRIS. It is worth noting that we can achieve superior performance to ETRIS only using the local intertwined module with much fewer tuning parameters, and more details are shown in Sec. 3.5.

Overall, BarLeRIa achieves a significant improvement of +3.7 IoU on average across the three RefCOCO-related datasets, demonstrating its superiority over existing RIS methods.

Table 2: Comparison with full fine-tuning SOTA RIS methods and these methods either utilize large language models or are pre-trained with additional datasets. IoU is utilized as the metric. $^\dagger$ denotes that the model is tuned using the mixed RefCOCO datasets.

| Method | RefCOCO | | | RefCOCO+ | | | G-Ref | | Avg |
|---|---|---|---|---|---|---|---|---|---|
| | val | testA | testB | val | testA | testB | val(u) | test(u) | |
| LISA-7B (Lai et al., 2023) | 74.1 | 76.5 | 71.1 | 62.4 | 67.4 | 56.5 | 66.4 | 68.5 | 67.9 |
| LISA-7B (ft) (Lai et al., 2023) | 74.9 | 79.1 | 72.3 | 65.1 | 70.8 | 58.1 | 67.9 | 70.6 | 69.9 |
| PolyFormer-B$^\dagger$ (Liu et al., 2023) | 74.8 | 76.6 | 71.1 | 67.6 | 59.3 | 72.9 | 67.8 | 69.1 | 69.9 |
| UNINEXT-R50$^\dagger$ (Yan et al., 2023) | **77.9** | **79.7** | **75.8** | 66.2 | 71.2 | 59.0 | 70.0 | 70.5 | 71.3 |
| ETRIS (Xu et al., 2023) | 72.4 | 74.6 | 69.3 | 64.5 | 70.4 | 56.9 | 62.6 | 63.1 | 66.1 |
| BarLeRIa | 75.0 | 77.1 | 71.2 | 68.6 | 73.2 | 61.2 | 65.9 | 66.4 | 69.8 |
| BarLeRIa-Mixed$^\dagger$ | 77.6 | 79.4 | 75.3 | **71.7** | **75.7** | **66.0** | **70.9** | **71.4** | **73.5** |

## 3.3 COMPARISON TO FULL FINE-TUNING METHODS

We further conducted a comparison between our proposed approach and existing SOTA full fine-tuning methods. These methods either utilize large language models or are pre-trained with additional datasets that contain region-level information. Without using additional datasets, we select a superior CLIP version, EVA-CLIP, which is still pre-trained using general-purpose datasets. For fair comparisons, we also use the EVA-CLIP pre-trained vision language model as the backbone for ETRIS. As shown in Tab. 2, BarLeRIa outperforms ETRIS by a significant margin, achieving an average improvement of +3.7 IoU using the same EVA-CLIP pre-trained backbone.

Compared to LISA-7B (Lai et al., 2023), a large vision language model with 7 billion parameters, our approach demonstrates a significant improvement when LISA-7B is not fine-tuned and achieves comparable performance when LISA-7B is fully fine-tuned. Compared with PolyFormer-B (Liu et al., 2023) that utilizes Swin-B (Liu et al., 2021b) as the visual encoder and the BERT transformer as the textual encoder, our proposed BarLeRIa achieves comparable performance without additional region-level pre-training and mixed fine-tuning. It is worth noting that PolyFormer introduces a second pre-training phase to incorporate region-level information using additional datasets, including Visual Genome, three RefCOCO-related datasets, and Flickr30k-entities. Furthermore, BarLeRIa achieves a +3.6 IoU improvement over PolyFormer-B when we additionally employ mixed fine-tuning. UNINEXT (Yan et al., 2023) leverages pre-training on Objects365 to learn region-level information and also employs mixed fine-tuning. BarLeRIa achieves a +2.2 IoU improvement over UNINEXT-R50 when we also employ mixed fine-tuning with much fewer tuning parameters.

We also conduct experiments using the ViT-Large visual encoder to verify the generalization ability of our method across different architectures. As shown in Tab. 3, BarLeRIa-L outperforms PolyFormer-L without additional region-level pre-training and mixed fine-tuning. Moreover, compared to the best-performing RIS method, UNINEXT, BarLeRIa-L-Mixed achieves a clear margin of +1.0 IoU averaged improvement across RefCOCO-related datasets, demonstrating its effectiveness.

## 3.4 VISUALIZATION

As illustrated in Fig. 3, we present visualization results with different settings under easy scenarios and hard scenarios, respectively. In the figure, (d) SNF means we just use normalizing flow to adapt the visual features without the bridge used in ETRIS, and (e) SNF+ETRIS means we combine SNF with ETRIS. We use these two settings to determine whether SNF is the key to PET RIS approaches. We find that both (d) and (e) lag much compared with our BarLeRIa and prove that our proposed two PET modules provide great improvement (more details of the ablation are shown in Sec. 3.5).

The first two rows of Fig. 3 represent the easy scenario and all methods can segment objects correctly. The difference is only in the detail and the finesse of the contours. BarLeRIa and BarLeRIa-L-mixed achieve the best segmentation IoU while ETRIS performs worst. For the hard scenario, *i.e.*, the last two rows of Fig. 3, ETRIS fails to locate the object correctly, SNF and SNF+ETRIS introduce overly large outlines, indicating that they do not fully understand the text description, while our BarLeRIa fully understands the meaning of the text and accurately segments the target objects.

Table 3: Comparison with full fine-tuning SOTA RIS methods using ViT-Large as the visual backbone. These methods either utilize large language models or are pre-trained with additional datasets. IoU is utilized as the metric. [†] denotes that the model is tuned using the mixed RefCOCO datasets.

| Method | RefCOCO | | | RefCOCO+ | | | G-Ref | | Avg |
|---|---|---|---|---|---|---|---|---|---|
| | val | testA | testB | val | testA | testB | val(u) | test(u) | |
| PolyFormer-L[†] (Liu et al., 2023) | 76.0 | 78.3 | 73.3 | 69.3 | 61.9 | 74.6 | 69.2 | 70.2 | 71.6 |
| UNINEXT-L[†] (Yan et al., 2023) | **80.3** | **82.6** | **77.8** | 70.0 | 74.9 | 62.6 | **73.4** | **73.7** | 74.4 |
| BarLeRIa-L | 76.8 | 79.0 | 74.0 | 71.5 | 76.2 | 65.4 | 68.7 | 69.7 | 72.7 |
| BarLeRIa-L-Mixed[†] | 79.0 | 80.8 | 77.0 | **74.2** | **77.8** | **68.3** | 72.7 | 73.3 | **75.4** |

Language: "kid running"

Language: "horse closest to us"

Language: "cut banana in bowl"

Language: "bear with face most showing"

(a) Image  (b) GT  (c) ETRIS  (d) SNF  (e) SNF+ETRIS  (f) BarLeRIa  (g) BarLeRIa-L*

Figure 3: Qualitative results with different settings. (a) the input image. (b) the ground truth. (c) ETRIS. (d) SNF without local intertwined module. (e) SNF+ETRIS. (f) our proposed BarLeRIa. (g) BarLeRIa-L using mixed datasets. Best viewed in color.

## 3.5 ABLATION STUDY

To establish the efficacy of our proposed approach, we perform ablation studies on the components of our proposed BarLeRIa. We only briefly document the averaged performance of different test splits for RefCOCO, RefCOCO+, and G-Ref respectively (please refer to the appendix B for detailed results). Illustrated in Tab. 4, SNF means we use the normalizing flow to adjust the feature, LIM is the abbreviation of Local Intertwined Module, GST denotes Global Shortcut Tuning, and No Global means we just use the Local Intertwined Module without the Global Shortcut Tuning. As we can see, just employing existing SNF or combining SNF with ETRIS does not improve segmentation performance. Besides, if we only use the local intertwined module (No Global in the table), we can outperform ETRIS with +2.6 averaged IoU improvement with nearly one-tenth the number of tuning parameters. This result demonstrates that BarLeRIa can greatly surpass existing PET state-of-the-art with fewer learnable parameters and showcases its superiority. Finally, with the proposed Global Shortcut Tuning, BarLeRIa achieves further enhancements to +1.1 averaged IoU.

## 4 RELATED WORK

**Parameter Efficient Tuning** (PET) adjust only a fraction of the parameters and alleviate the computational challenges associated with fine-tuning the entire model. One prominent research direction focuses on incorporating lightweight architectures into the frozen backbone and updating only these newly added architectures during fine-tuning (Houlsby et al., 2019; Mahabadi et al., 2021; Lester

Table 4: Ablation study on the components of BarLeRIa. LIM is the abbreviation of Local Intertwined Module, GST denotes Global Shortcut Tuning, and No Global means we just use the Local Intertwined Module without the Global Shortcut Tuning.

| Method | SNF | LIM | GST | Params(M) | RefCOCO | RefCOCO+ | G-Ref | Avg |
|---|---|---|---|---|---|---|---|---|
| ReSTR | - | - | - | 86.19 | 67.0 | 54.8 | 54.5 | 58.8 |
| ETRIS | × | × | × | 1.39 | 70.2 | 59.1 | 59.2 | 62.8 |
| SNF | ✓ | × | × | **0.18** | 70.6 | 59.6 | 59.3 | 63.2 |
| SNF+ETRIS | ✓ | × | × | 1.57 | 70.2 | 59.9 | 60.1 | 63.4 |
| No Global | ✓ | ✓ | × | 0.39 | 71.4 | 63.1 | 61.6 | 65.4 |
| BarLeRIa | ✓ | ✓ | ✓ | 2.21 | **72.2** | **64.2** | **62.9** | **66.5** |

et al., 2021; Li & Liang, 2021; Karimi Mahabadi et al., 2021; Chen et al., 2022b; Jie & Deng, 2022; Jia et al., 2022). For instance, AdaptFormer (Chen et al., 2022b) and ConvPass (Jie & Deng, 2022) introduce bottleneck or convolution modules along the skip connections within transformer layers and adapt the residuals for downstream tasks. Recently, Wang et al. (2023) proposed leveraging normalizing flows to adjust the shortcuts rather than the residuals within transformer layers, offering an easily implementable and accessible approach for various architectures. Another line of the PET method involves updating only a subset of the parameters in the original model (Sung et al., 2021; Zaken et al., 2021). Zaken et al. (Zaken et al., 2021), for example, demonstrate that updating only the bias terms can achieve competitive or even superior performance compared to full fine-tuning. Additionally, some researchers have explored matrix decomposition techniques to reduce the number of learnable parameters by factorizing the weights of pre-trained models (Hu et al., 2021; Jie & Deng, 2023), which also yield satisfactory performance. Unfortunately, PET approaches for referring image segmentation are less investigated. Recently, Xu et al. (2023) introduced PET to referring image segmentation by leveraging the bridge module for information fusion between visual and textual modalities. However, their proposed ETRIS lacks feature adaption and global visual regularization, resulting in unsatisfactory performance. Besides these works, some researchers factorize weights of the pre-trained model based on the low-rank assumption, such that parameters that need to be tuned can be largely reduced Hu et al. (2021).

**Referring Image Segmentation** (RIS) aims to segment a target instance or region referred by the given text query and is initially introduced by Hu et al. (2016). Early methods were predominantly based on the CNN+LSTM approach (Liu et al., 2017; Li et al., 2018), where the image and text inputs were encoded separately using their respective backbones. However, in recent years, transformer architectures have gained popularity due to their flexibility and scalability(Vaswani et al., 2017; Dosovitskiy et al., 2020), allowing RIS methods to employ a unified architecture across different modalities (Kim et al., 2022; Yang et al., 2021; Liu et al., 2023; Yan et al., 2023). Additionally, the advent of multi-modal pre-training (Radford et al., 2021) has provided RIS models with the advantage of leveraging large-scale pre-training data (Wang et al., 2022). Besides, recent work (Cheng et al., 2023) has proven that the global prior can help the referring segmentation. However, these methods require full fine-tuning of an additional over-parameterized model and divide the segmentation process into two stages without end-to-end training.

## 5 CONCLUSION

In this paper, we pay attention to parameter efficient tuning for referring image segemntation. We reveal that previous approaches focus on vision and language modal alignment, but ignores adapting the biased feature from pre-trained models. Besides, previous approaches fuse the local visual features with the textual embeddings without introducing global prior from text input to regularize the visual feature. To address these issues, we propose a novel PET framework BarLeRIa: Bi-directional Intertwined Vision Language Efficient Tuning for Referring Image Segmentation , which leverages intertwined vision language adapters and bi-directional tuning framework to fully exploit the frozen pre-trained models' potential. We conduct extensive experiments on three RefCOCO-related benchmarks. BarLeRIa consistently outperforms prior parameter efficient tuning methods with a clear margin. Moreover, BarLeRIa also surpasses full fine-tuning state-of-the-art approaches without pre-training using additional training datasets.

**Acknowledgment** This work was supported in part by the National Natural Science Foundation of China under Grant 62125109, Grant 62250055, Grant 61931023, Grant 61932022, Grant 62371288, Grant 62320106003, Grant 62301299, Grant T2122024, Grant 62120106007.

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

# A  EXPERIMENTAL SETUP

## A.1  DATASETS

**RefCOCO** (Kazemzadeh et al., 2014) is a widely used benchmark for referring image segmentation out of its large scale. It consists of 19,994 images with annotations for 142,210 referring expressions related to 50,000 objects. These annotations were gathered through a two-player game based on the MSCOCO dataset. The dataset is divided into four subsets: 120,624 samples for training, 10,834 for validation, 5,657 for test A, and 5,095 for test B. On average, the referring expressions are 3.6 words long, and each image contains at least two objects.

**RefCOCO+** (Kazemzadeh et al., 2014) contains 141,564 referring expressions associated with 49,856 objects across 19,992 images. It is designed to be more challenging than RefCOCO by excluding certain absolute-location words. Similar to RefCOCO, RefCOCO+ is divided into four subsets: 120,624 samples for training, 10,758 for validation, 5,726 for test A, and 4,889 for test B.

**G-Ref** (Yu et al., 2016) consists of 104,560 referring expressions related to 54,822 objects in 26,711 images. The expressions in G-Ref were collected from Amazon Mechanical Turk and have an average length of 8.4 words. Compared with the previous two datasets, G-Ref contains more descriptions of locations and appearances in the text annotations. Note that we apply both the Google and UMD split in our experiments with the denotation G-Ref(g) and G-Ref(u) respectively.

Language: "it all looks yummy"

Language: "part of table without anything on it"

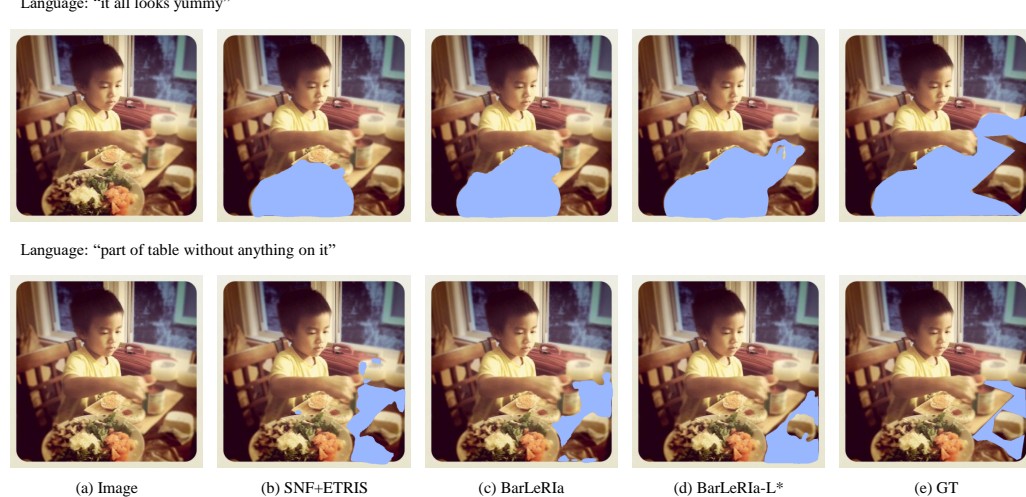

| (a) Image | (b) SNF+ETRIS | (c) BarLeRIa | (d) BarLeRIa-L* | (e) GT |

Figure 4: Visualization results with different settings. (a) the input image. (b) SNF+ETRIS. (c) BarLeRIa. (d) BarLeRIa-L using mixed datasets. (e) the ground truth. Best viewed in color.

Language: "older sheep"

Language: "small sheep towards us"

Language: "largest sheep"

Language: "closest little one"

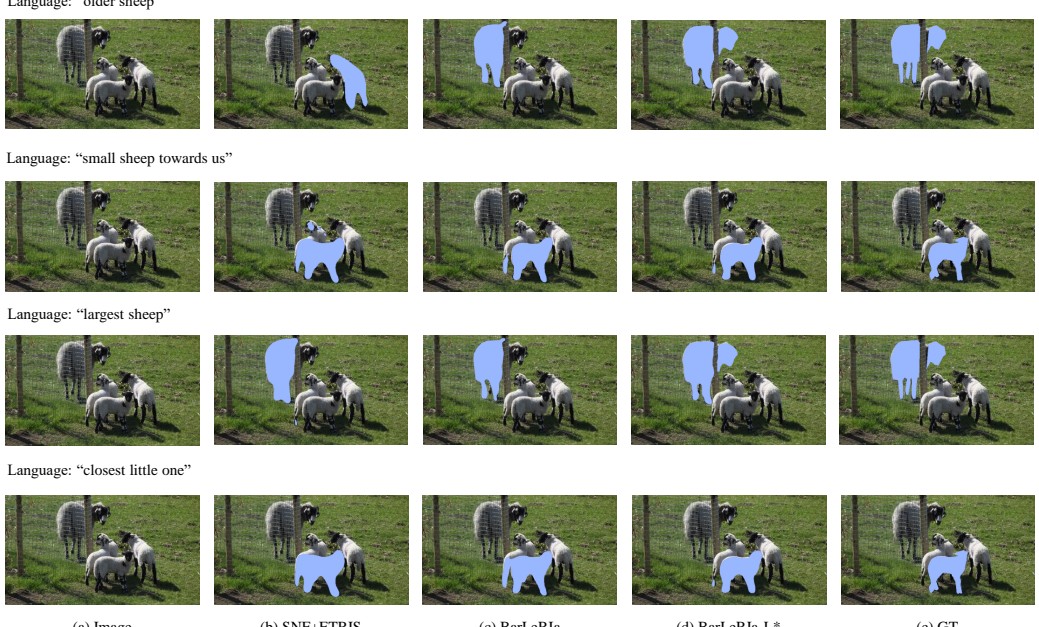

| (a) Image | (b) SNF+ETRIS | (c) BarLeRIa | (d) BarLeRIa-L* | (e) GT |

Figure 5: Visualization results with different settings. (a) the input image. (b) SNF+ETRIS. (c) BarLeRIa. (d) BarLeRIa-L using mixed datasets. (e) the ground truth. Best viewed in color.

## A.2 IMPLEMENTATION DETAILS

We choose ViT-B/16 of CLIP as the pretrained vision language model in most experiments. Following Wang et al. (2022), we resize the input images to $416 \times 416$. For the visual backbone and textual backbone, we follow the settings of CRIS (Wang et al., 2022) and ETRIS (Xu et al., 2023). For the local intertwined module, we employ SNF-deep, which has four transformations in MHSA block and MLP block respectively, in each visual transformer layer and text transformer layer. Following Wang et al. (2023), the explicit log Jacobian loss is not used when calculating the training objective. For the global shortcut tuning network that consists of 4 global shortcut tuning modules. We set the feature dimension 144 as a default setting. We employ 128 learnable query tokens following VPT . The output 4 global shortcut features are fused with the output from the 3, 6, 9, 12 visual encoder layers, respectively. The referring image segmentation head consists of a

Table 5: The detailed ablation study on the components of BarLeRIa.

| Method | Params(M) | RefCOCO | | | RefCOCO+ | | | G-Ref | | | Avg |
| --- | --- | --- | --- | --- | --- | --- | --- | --- | --- | --- | --- |
| | | val | testA | testB | val | testA | testB | val(u) | test(u) | val(g) | |
| ReSTR | 86.19 | 67.2 | 69.3 | 64.5 | 55.8 | 60.4 | 48.3 | 54.5 | - | 54.5 | 58.8 |
| ETRIS | 1.39 | 70.5 | 73.5 | 66.6 | 60.1 | 66.9 | 50.2 | 59.8 | 59.9 | 57.9 | 62.8 |
| SNF | 0.18 | 70.8 | 73.8 | 67.2 | 60.5 | 67.1 | 51.1 | 60.0 | 60.0 | 58.0 | 63.2 |
| SNF+ETRIS | 1.57 | 70.4 | 73.3 | 66.9 | 60.5 | 66.1 | 53.0 | 60.6 | 60.8 | 59.0 | 63.4 |
| No Global | 0.39 | 71.7 | 74.6 | 68.0 | 64.0 | 70.2 | 55.1 | 62.2 | 62.3 | 60.2 | 65.4 |
| BarLeRIa | 2.21 | 72.4 | 75.9 | 68.3 | 65.0 | 70.8 | 56.9 | 63.4 | 63.8 | 61.6 | 66.5 |

transformer decoder and an up-sampling projector, and all the settings follow Wang et al. (2022). We train the whole network in an end-to-end manner for 50 epochs using the Adam optimizer with a learning rate of 0.0001. A learning rate decay is employed at the 35th epoch with a decay factor of 0.1. We train the model using 2 Tesla V100 GPUs with a batch size of 32. For ViT-L/14, we train the model using 8 Tesla V100 GPUs with a batch size of 64 and an initial learning rate of 0.0002. Following previous works (Ding et al., 2021; Liu et al., 2017; Wang et al., 2022; Xu et al., 2023), we adopt IoU as the metric to evaluate the performance. The IoU calculates intersection regions over union regions of the predicted segmentation mask and the ground truth. Given the cross-modal intertwined feature $F_{ci}$ and the transformed textual feature $F_t$, the segmentation result is obtained by reshaping $\sigma(F_{ci} \cdot F_t)$ into $\frac{H}{4} \times \frac{W}{4}$ and then up-sampling it back to the original image size, following Wang et al. (2022), where $\sigma$ denotes the sigmoid function, $H$ and $W$ are the origin shape of the input images.

# B  MORE EXPERIMENTAL RESULTS

## B.1  DETAILED ABLATION

To establish the efficacy of our proposed approach, we perform ablation studies on the components of our proposed BarLeRIa. In this detailed ablation, we document all the performance for different splits respectively. Illustrated in Tab. 5, SNF means we use the normalizing flow to adjust the feature, LIM is the abbreviation of Local Intertwined Module, GST denotes Global Shortcut Tuning, and No Global means we just use the Local Intertwined Module without the Global Shortcut Tuning. As we can see, just employing existing SNF or combining SNF with ETRIS does not improve segmentation performance. Besides, if we only use the local intertwined module (No Global in the table), we can outperform ETRIS with +2.6 averaged IoU improvement with nearly one-tenth the number of tuning parameters. This result demonstrates that BarLeRIa can greatly surpass existing PET state-of-the-art with fewer learnable parameters and showcases its superiority. Finally, with the proposed Global Shortcut Tuning, BarLeRIa achieves further enhancements to +1.1 averaged IoU.

## B.2  MORE VISUALIZATION

As illustrated in Fig. 4, 5, 6, and 7 we present visualization results with different settings. In the figure, (b) SNF+ETRIS means we combine SNF with ETRIS, and (d)) BarLeRIa-L* denotes we use ViT-large as the visual backbone and employ mixed datasets tuning using BarLeRIa. All the results demonstrate that our proposed intertwined vision language tuning algorithm and Bi-directional efficient tuning framework successfully adapt the features, introduce a global regularization, and yield a significant segmentation performance improvement.

# C  THE DETAILS OF THE REFERRING IMAGE SEGMENTATION HEAD

As we elaborated in Sec. 2.5, we follow Wang et al. (2022) and Xu et al. (2023), and incorporate a learnable referring image segmentation head composed of a cross-modal neck, vision-language decoder, and an up-sample projector to extract the cross-modal intertwined feature and the transformed

Language: "giraffe half way from fence"

Language: "girafe middle height"

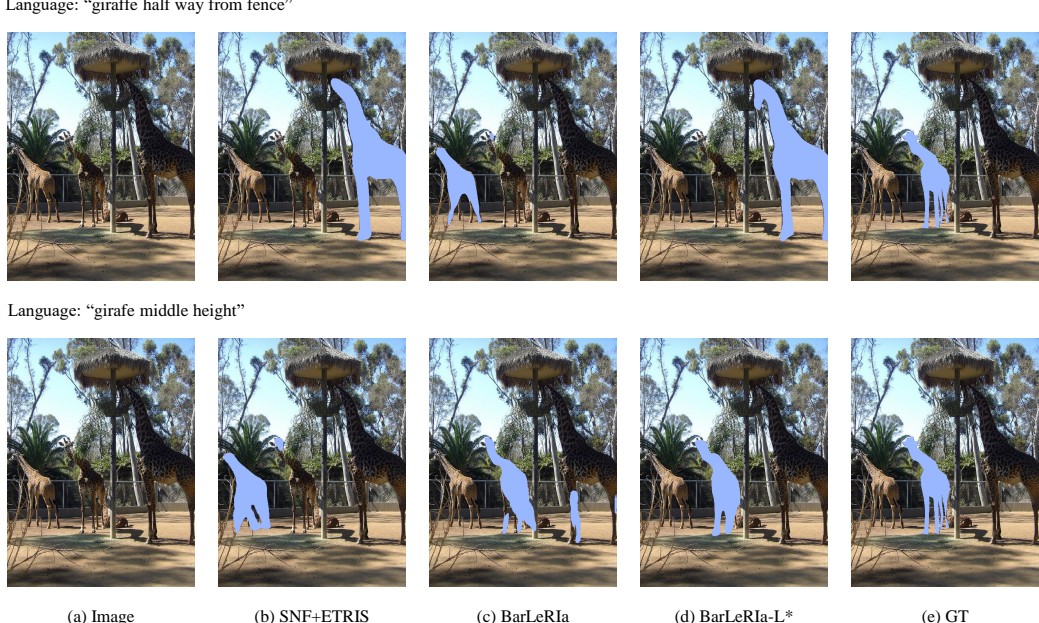

(a) Image      (b) SNF+ETRIS      (c) BarLeRIa      (d) BarLeRIa-L*      (e) GT

Figure 6: Visualization results with different settings. (a) the input image. (b) SNF+ETRIS. (c) BarLeRIa. (d) BarLeRIa-L using mixed datasets. (e) the ground truth. Best viewed in color.

Language: "donut no hole"

Language: "no hold pastry"

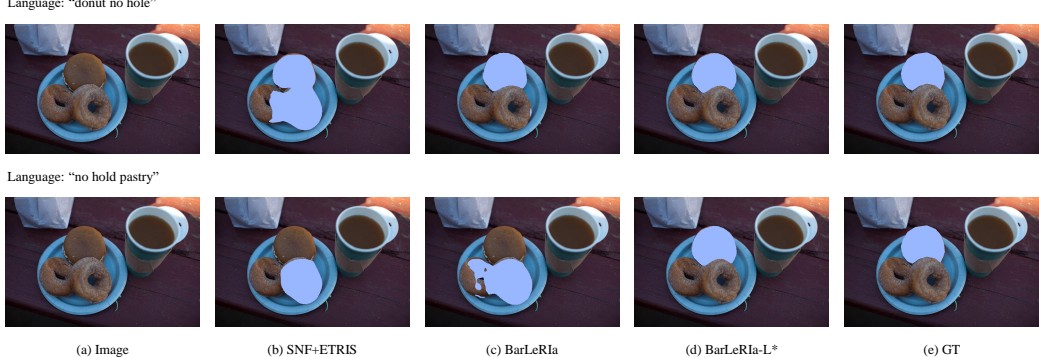

(a) Image      (b) SNF+ETRIS      (c) BarLeRIa      (d) BarLeRIa-L*      (e) GT

Figure 7: Visualization results with different settings. (a) the input image. (b) SNF+ETRIS. (c) BarLeRIa. (d) BarLeRIa-L using mixed datasets. (e) the ground truth. Best viewed in color.

textual feature. Here we introduce the referring image segmentation head in detail.

**Cross-modal Neck:** Following Xu et al. (2023), given multiple adapted visual features $F_v^i, i \in 2, \cdots, N$ from different stages and the adapted textual embeddings $F_l$, we employ MHSA with convolution to obtain the fusion features $F_f$. Then, we concatenate a 2D spatial coordinate feature $F_{coord}$ with the fused features $F_f$ and use a $3 \times 3$ convolution to further fuse them as:

$$F_c = Conv([F_f, F_{coord}]) \tag{11}$$

**Vision-Language Decoder:** Then, we add the feature $F_c$ outputted from the cross-modal neck and the output from the last global shortcut tuning module $F_{out}^M$ as $F_c + F_{out}^M$ and input the combined features along with textual embeddings $F_l$ to a vision-language decoder following Wang et al. (2022); Xu et al. (2023) to achieve the multi-modal features $F_{mm}$. Specifically, the decoder consists of three layers, with each layer composed of a multi-head self-attention layer, a multi-head cross-attention layer, and a feed-forward network. Within each decoder layer, the input combined feature is initially passed through the multi-head self-attention layer to capture global contextual information. Subsequently, the cross-attention layer is employed to facilitate multi-modal interaction, enabling the transmission of detailed semantic information from the textual features to the visual features. This interaction is achieved by mapping visual features to queries and textual features to keys and

values. Following the cross-attention layer, an MLP block comprising two layers, along with Layer Normalization and residual connections, is utilized to further process the output features. Finally, the output of the MLP block is employed to generate the final segmentation mask.

**Up-sample Projector:** To obtain mask prediction on each pixel according to the corresponding semantic information, we follow Wang et al. (2022); Xu et al. (2023) use a projector to make transformations on multi-modal features $F_{mm}$ and sentence-level feature $F_l$ to extract the cross-modal intertwined feature $F_{ci}$ and the transformed textual feature $F_t$ as:

$$F_{ci} = Conv(Up(F_{mm}))F_t = Linear(F_l) \tag{12}$$

where UpSample denotes 4× upsampling, and convolution and linear projection are used to transform $F_{mm}$ and $F_l$ into a suitable dimension following Wang et al. (2022); Xu et al. (2023).

## D  LIMITATION AND FUTURE WORK

Despite its ability to achieve state-of-the-art performance on referring image segmentation, Bar-LeRIa has some limitations. The proposed global shortcut tuning network is relatively larger compared with our local intertwined vision language tuning module (1.8M v.s. 0.4M). A more efficient approach that imposes global prior back to the visual features without influencing the original feature adaption deserves future research. Besides, because it involves the processing of variable-length sequences, our approach is not compatible with convolutional networks. How to adapt the convolutional neural network to make our method more general also deserves further research. Finally, open-vocabulary zero-shot referring image segmentation is a worthwhile direction to explore as multi-modal large-scale models continue to evolve.

