# OpenReview forum: "BarLeRIa: An Efficient Tuning Framework for Referring Image Segmentation"
_ICLR.cc/2024/Conference — ICLR 2024 spotlight_

### Official Review · Reviewer_8Qir · 2023-10-31

**Soundness:** 3 good
**Presentation:** 2 fair
**Contribution:** 3 good
**Rating:** 8
**Confidence:** 4

**Summary:**

This paper proposes a novel Parameter-Efficient Tuning framework for Referring Image Segmentation. By designing a bi-directional intertwined vision-language adapter, this frame exploits the frozen power of pre-trained vision-language models. In addition, this paper introduces a Global Prior Module and a Global Shortcut Tuning Network to extract global prior from text input to regularize the visual feature.
Combining these contributions, this paper outperforms previous parameter-efficient tuning methods on RIS benchmarks and even surpasses SOTA full fine-tuning approaches on several tasks of RIS benchmarks.

**Strengths:**

* Strong performance on three referring image segmentation benchmarks.
* The well-designed Local Intertwined Module: according to the Table 4: Local Intertwined Module improves can outperform ETRIS with very small amount parameters.

**Weaknesses:**

* The limited improvement brought by Global Shortcut Tuning.  According to Table 5, Compared with No Global, BarLeRIa achieves 72.2 on RefCOCO using 2.21M, while without GST, the model achieves 71.4 using 0.39M. It seems that the model without GST owns a better balance between performance and parameters.

**Questions:**

* See in Weaknesses.
* Is any possible to provide the training time comparison between ETRIS and BarLeRIa?

---

> ### Author Response · Authors · 2023-11-17
> **Response to Reviewer 8Qir**
>
> We are grateful for your time in reviewing this article and appreciate your recognition of our contribution. Below, we provide answers to each question.
>
> - **Q1: The limited improvement brought by Global Shortcut Tuning**
>
>   **A1:** Thank you for your meticulous review! The improvement brought by Global Shortcut Tuning is not limited. The reason is fourfold.
>   1) Compared to No Global, Global Shortcut Tuning achieves an average improvement of +1.1 mIou, which is noteworthy given that No Global only achieves 65.4 average mIou.
>   2) Compared to the fine-tuned baseline ReSTR[1], which requires 86.2 M parameter updates, BarLeRIa updates 2.21 M parameters, which is still very lightweight.
>   3) Local Intertwined Module and Global Shortcut Tuning enhance referring image segmentation from two different perspectives both in a parameter-efficient manner.
>   No Global's superiority over ETRIS (0.39M parameters) demonstrates our proposed local intertwined module's success in feature alignment and adjustment, addressing PET's core feature tuning issue, previously overlooked by ETRIS. However, the local intertwined module lacks global prior from textual embeddings, and an efficient global regularization method is still needed for Referring Image Segmentation.
>   4) While existing WiCo[2] proposes to utilize global prior along with local feature fusion for referring image segmentation, it requires an additional ResNet-50 model (25M parameters), which is heavy. In contrast, our proposed Global Shortcut Tuning incorporates global regularization in a parameter-efficient manner, making it efficient.
>
>   [1] Namyup Kim, Dongwon Kim, Cuiling Lan,Wenjun Zeng, and Suha Kwak. Restr: Convolution-free referring image segmentation using transformers. In Proceedings of the IEEE/CVF Conference on Computer Vision and Pattern Recognition, pp. 18145–18154, 2022.
>
>   [2] Zesen Cheng, Peng Jin, Hao Li, Kehan Li, Siheng Li, Xiangyang Ji, Chang Liu, and Jie Chen. Wico: Win-win cooperation of bottom-up and top-down referring image segmentation. arXiv preprint arXiv:2306.10750, 2023.
>
> - **Q2: Training time between ETRIS and BarLeRIa**
>
>   **A1:** Thanks for your suggestion.
>   As shown in the table below, we experiment with Refcoco+ on 2 Tesla V100 GPUs with a batch size of 32, as described in the implementation details, and record the average one-epoch training time. BarLeRIa and No Global are the fastest since they don't require training extra cross attention for feature fusion, indicating that our proposed local intertwined module is a more efficient way to align and adjust features.
>   |    Method   |Trainig time (s) |Avg Performance
>   | :---        |    :----:   |          :---:
>   |ETRIS  |1230  |  62.8
>   | ETRIS+SNF| 1779   | 63.4
>   | No Global | 1032   | 65.4
>   | BarLeRIa |   1087  | 66.5

---

> > ### Comment · Reviewer_8Qir · 2023-11-23
> > **Reply to authors**
> >
> > Thank you for your kind response. Your reply addresse my inquiries. I also review the queries raised by other reviewers, and find their remarks insightful. Based on your feedback and the suggestions of others, I will improve the rating to 7.

---

### Official Review · Reviewer_eHeC · 2023-11-01

**Soundness:** 4 excellent
**Presentation:** 3 good
**Contribution:** 3 good
**Rating:** 8
**Confidence:** 4

**Summary:**

This paper considers the problem of using parameter efficient fine-tuning in referring image segmentation (RIS). A sophisticated PEFT paradigm is proposed, consisting of Adapting Shortcut with Normalizing Flow (SNF),  Local Intertwined Module (LIM), and Global Shortcut Tuning (GST). Results are presented on commonly used RIS benchmarks of RefCOCO, RefCOCO+, and G-Ref. A detailed evaluation shows superior results of the proposed method over previous PEFT method in RIS.

**Strengths:**

- The paper is well written. Adequate background is provided, the literature review covers very recent approaches, and the paper provides necessary background to the required concepts.
- It seems novel to me to use a small side network that takes text+image features with attention and output side features containing global priors. And the ablation study shows a significant improvement with this module.
- The proposed approach gets strong results that beats the previous parameter efficient fine-tuning approach in RIS.
- The paper conducts a thorough ablation study to show the effectiveness of each proposed module.

**Weaknesses:**

- Writing in Section 3.3 and 3.4 is not clear enough. It should be possible to write in one equation for (2)(3)(4) with a better subscription. Also it's a bit unclear whether $F_v^i$ is the same as (the $i^{th}$) $[cls, embed]$.
- (typo): it should be intertwine not interwine in Figure 2.

**Questions:**

Does the side network (Global Shortcut Tuning Network) not operate in parallel with the vision/language encoder? And is it required to cache each $F_v^i$ for the side network? If so, what is the extra time and memory cost for this approximately?

---

> ### Author Response · Authors · 2023-11-17
> **Response to Reviewer eHeC**
>
> We sincerely appreciate the time you have taken to review our article. We are also thankful for your recognition of our contribution. Below, we provide detailed answers to each of the questions you raised.
>
> - **Q1: Writing in Sections 3.3 and 3.4**
>
>   **A1:** Thanks for your suggestion, a unified subscription for equation (2), (3), and (4) will make the approach clearer. We will revise Sections 3.3 and 3.4 in the revised manuscript;
>   Yes, $F_v^i$ is the same as (the $i^{th}$) $[cls,embed]$. We are sorry for the unclear description.
>
>
> - **Q2: Typos**
>
>   **A2:** Thank you very much for your detailed review, we will fix all the typos in the new version!
>
> - **Q3: Cost of the proposed Global Shortcut Tuning Network**
>
>   **A3:**
>   Thanks for your meaningful question.
>   We conducted experiments on Refcoco+ using 2 Tesla V100 GPUs with a batch size of 32, as described in the implementation details. We recorded the averaged one-epoch training time and GPU memory cost for four approaches. Additionally, we evaluated the model using the Refcoco+ val datasets and recorded the whole test time and performance.
>   |    Method   |Trainig time (s) | Memory cost (G)| Inference time (s) | Performance
>   | :---        |    :----:   |          :---: |          :---: |          :---:
>   | ETRIS  |  1230| 32.2| 626 | 60.1
>   | ETRIS+SNF| 1779 | 35.4  |  701 | 60.5
>   | No Global | 1032 | 35.5  | 644 | 64.0
>   | BarLeRIa |   1087  |41.3 | 727 | 65.0
>
>   We analyze the results from three perspectives.
>
>   1. In terms of training speed, No Global and BarLeRIa are the fastest as they do not require extra cross attention for feature fusion, unlike ETRIS. This highlights the efficiency of our proposed local intertwined module as a feature alignment and adjustment method.
>   2. Regarding training memory cost, Global Shortcut Tuning Network requires the global prior from the visual backbone, which entails caching intermediate visual feature maps for 4 Global Shortcut Tuning blocks. This requires additional memory, but it is still more efficient than the previous method[1], which needed an additional ResNet-50 model. The mere addition of 6.8G of GPU memory is thus acceptable and efficient.
>   3. For the inference speed, BarLeRIa achieves a comparable speed to ETRIS+SNF while offering a +3.5 mIou performance improvement. Although still slightly slower than ETRIS and No Global, BarLeRIa adds global constraints to Referring Image Segmentation at a fraction of the cost in a parameter-efficient tuning manner. This would otherwise require an additional 25M ResNet model[1] and would not be able to be trained end-to-end.
>
>   [1] Zesen Cheng, Peng Jin, Hao Li, Kehan Li, Siheng Li, Xiangyang Ji, Chang Liu, and Jie Chen. Wico: Win-win cooperation of bottom-up and top-down referring image segmentation. arXiv preprint arXiv:2306.10750, 2023.

---

### Official Review · Reviewer_iAN9 · 2023-11-01

**Soundness:** 3 good
**Presentation:** 3 good
**Contribution:** 3 good
**Rating:** 6
**Confidence:** 2

**Summary:**

The authors have proposed a novel intertwined vision language efficient tuning algorithm based on the large-scale CLIP model. They claim that the previous methods overlook adapting the biased feature from pre-trained models and global prior regularization. The proposed method achieves state-of-the-art performance on RefCOCO, RefCOCO+, and G-Ref.

**Strengths:**

1. The proposed method brings a few trainable parameters into CLIP for both feature adaptation and modal fusion, which achieves the best performance on several RIS datasets.
2. The proposed method utilizes a novel intertwined structure to assist modal feature fusion.

**Weaknesses:**

1. The global prior is predicted after the vision-language blocks. Then, the global prior is input to the global shortcut tuning module. However, such a design is quite a long path. The inference speed may be very slow.
2. The comparison with other methods may be unfair. The proposed method utilizes ViT-Large as its backbone.
3. The dimensions and meanings of different symbols are not introduced in the paper, such as [Fl, Fv].
4. The decoder is not introduced in this paper. Although the authors have claimed they follow previous works, it is not detailed enough.

**Questions:**

Could you provide any results of the inference time compared with other parameter tuning methods?

---

> ### Author Response · Authors · 2023-11-17
> **Response to Reviewer iAN9**
>
> We sincerely appreciate you taking the time to review this article. Thank you also for recognizing our contribution. The answers to each question are provided below.
>
> - **Q1: The inference speed**
>
>   **A1:**
>   Thanks for your suggestion! We conduct experiments on the Refcoco+ dataset using two Tesla V100 GPUs with a batch size of 32 (as described in the implementation details) and record the average training time per epoch for four approaches. Then, we evaluate the model using Refcoco+ val datasets and record the whole inference time and the performance.
>   The average training time per epoch and the inference time for each approach are shown below.
>   |    Method   |Trainig time (s) | Inference time (s) | Performance
>   | :---        |    :----:   |          :---: |          :---:
>   | ETRIS  |  1230| 626 | 60.1
>   | ETRIS+SNF| 1779   |  701 | 60.5
>   | No Global | 1032   | 644 | 64.0
>   | BarLeRIa |   1087   | 727 | 65.0
>
>   Based on the table, it appears that BarLeRIa's inference speed is slower than No Global's. This is because BarLeRIa's global shortcut tuning module requires regularization with the prior obtained from the backbone, which may need more computational resources. However, BarLeRIa's slowness is limited compared to ETRIS, with only a 16% inference speed loss resulting in a 4.9 mIou improvement.
>
>   Moreover, BarLeRIa is more lightweight and efficient compared to the previous approach[1], which requires an additional ResNet-50 to implement global information extraction.
>
>   Last but not least, BarLeRIa's training speed is faster compared to ETRIS because the Local Intertwined Module eliminates the need for additional cross-attention for feature fusion. In contrast, ETRIS+SNF has the slowest training speed since it requires tuning both language and vision backbones using SNF, as well as training additional cross-attentions.
>
>   [1] Zesen Cheng, Peng Jin, Hao Li, Kehan Li, Siheng Li, Xiangyang Ji, Chang Liu, and Jie Chen. Wico: Win-win cooperation of bottom-up and top-down referring image segmentation. arXiv preprint arXiv:2306.10750, 2023.
>
> - **Q2: The comparison with other methods**
>
>   **A2:**  We apologize for any confusion that may arise from our experimental setting. As detailed in Section 4.2, our proposed BarLeRIa leverages the same CLIP pre-trained vision language model of ViT-B/16 as employed in ETRIS, and exhibits a substantial performance enhancement, as evidenced in Table 1.
>
>   To ensure a fair and extensive comparison with existing full fine-tuning state-of-the-art (SOTA) methods, we also adopt ViT-Large as the visual backbone in our experiments, similar to PolyFormer-L and UNINEXT-L.
>   The results in Table 3 demonstrate that BarLeRIa outperforms existing full fine-tuning SOTA methods, indicating the effectiveness of our approach in leveraging the large-scale pre-training of CLIP for downstream tasks.
>
>   Thus, our comparison with other methods is fair and extensive.
>
> - **Q3: The dimensions and meanings of [Fl, Fv]**
>
>   **A3:** Thanks for your suggestion. In the main text, we have explained the meanings of Fl: the output language embeddings,
>   and Fv: the output vision feature.
>   The dimension for $F_l$ is $F_l\in\mathbb{R}^{B\times N_1\times C}$, where B is the batch size, $N_1$ is the length of word
>   sequence, and C is the feature dimension.
>   The dimension for $F_v$ is $F_v\in\mathbb{R}^{B\times N_2\times C}$, where B is the batch size, $N_2$ is the number of
>   image patches, and C is the feature dimension.
>
> - **Q4: The details of the decoder**
>
>   **A4:** In section 3.5, we have claimed that following [1,2], "we incorporate a learnable referring image segmentation head composed of a cross-modal neck, vision-language decoder, and an up-sample projector."
>   Given that the primary focus of this paper is on the bi-directional efficient tuning framework, we have not provided a detailed description of the decoder in the main body of the paper.
>   However, we appreciate your suggestion and will include a detailed discussion of the vision-language decoder settings in the appendix for the sake of completeness and clarity. Thank you!
>
>   [1] Zhaoqing Wang, Yu Lu, Qiang Li, Xunqiang Tao, Yandong Guo, Mingming Gong, and Tongliang Liu. Cris: Clip-driven referring image segmentation. In Proceedings of the IEEE/CVF conference on computer vision and pattern recognition, pp. 11686–11695, 2022.
>
>   [2] Zunnan Xu, Zhihong Chen, Yong Zhang, Yibing Song, Xiang Wan, and Guanbin Li. Bridging vision and language encoders: Parameter-efficient tuning for referring image segmentation. In Proceedings of the IEEE/CVF International Conference on Computer Vision, 2023.

---

### Meta-Review · Area_Chair_ZFH5 · 2023-12-06

**Metareview:**

The paper proposes a novel parameter-efficient tuning method with pretrained vision-language models for referring image segmentation. The proposed framework exploits the power of pretrained vision-language models using a bi-directional intertwined adapter. Also, the paper introduces a global prior module and a global shortcut tuning network to incorporate global prior from the text input for regularizing visual features. With the two techniques, the proposed method outperforms existing tuning methods on RIS. The reviewers unanimously acknowledge and appreciate the contributions of the work, leading the AC to recommend acceptance. While one reviewer initially raised concerns about experiment fairness and inference speed, these issues appear to have been addressed, although the reviewer has not provided further feedback.

**Justification For Why Not Higher Score:**

Despite its novelty and strong performance gains acknowledged by the reviewers, the proposed architecture is task-specific. As a result,
 the impact on the representation learning community may be limited.

**Justification For Why Not Lower Score:**

All the reviewers appreciate the value of the paper and most of the initial concerns are resolved by the reviewers resulting in a score at the top tail of the distribution.

---

### Decision · Program_Chairs · 2024-01-16

Accept (spotlight)